# Recognition of Community Pharmacists’ Behaviors Related to Information Sharing: A Cross-Sectional Study

**DOI:** 10.3390/pharmacy12020063

**Published:** 2024-04-06

**Authors:** Ryota Kumaki, Chika Kiyozuka, Mika Naganuma, Satoshi Yuge, Ryota Tsukioka, Hidehiko Sakurai, Keiko Kishimoto

**Affiliations:** 1Department of Social Pharmacy, Showa University Graduate School of Pharmacy, Tokyo 142-8555, Japankishimoto-k@pharm.showa-u.ac.jp (K.K.); 2Kraft Inc., Tokyo 100-8225, Japan; 3Qol Co., Ltd., Tokyo 105-8452, Japan; 4Nihon Chouzai Co., Ltd., Tokyo 100-6737, Japan; 5AIN Holdings Inc., Tokyo 151-0053, Japan; 6Faculty of Pharmaceutical Sciences, Hokkaido University of Science, Sapporo 006-8585, Japan

**Keywords:** community pharmacists, information sharing, pharmacists’ behavior, gap in recognition, trust in pharmacists, willingness to self-disclose

## Abstract

With the recent shift in community pharmacist services toward in-person services and the growing need for centralized and continuous medication management/monitoring, pharmacist–patient information sharing is crucial. This study investigated the pharmacist–patient gap in the recognition of pharmacists’ behaviors regarding information sharing and assessed the potential impact of such recognition on patient trust and willingness to self-disclose. This cross-sectional study included 600 patients (aged 21–85 years) using pharmacy services (surveyed online in December 2020) and 591 community pharmacists with ≥1 year of experience (surveyed from September to November 2021). Both groups responded to items on the recognition of pharmacists’ behaviors regarding information sharing. There were patient-specific items on trust in community pharmacists and willingness to self-disclose. For all items on the recognition of pharmacists’ behaviors, patients’ scores were significantly lower (4–5) than pharmacists’ own scores (≥5), revealing a notable perception gap. Patients’ recognition had a positive, direct effect on trust and willingness, and trust had a positive, direct effect on willingness. Patients’ recognition and trust positively influenced their willingness to self-disclose. Pharmacist communication with clear intent is important to bridge the gaps in pharmacist–patient recognition and foster effective patient–pharmacist relationships.

## 1. Introduction

Important aspects of community pharmacy services include the effective assessment of medication therapy, consideration of side effects, and improvement of medication adherence. In 2003, the U.S. Congress approved the Medicare Prescription Drug, Improvement, and Modernization Act [1], which facilitated various medication therapy management services [2,3,4]. In Japan, the 2015 formulation of the “Pharmacy Vision for Patients” [5] and revisions to dispensing fees [6] promoted a shift from in-object to in-person services, requiring centralized and continuous monitoring and pharmacological management of medications. However, it continues to be difficult for community pharmacists in Japan to check doctors’ medical records, nursing records, and laboratory values, causing them to rely mainly on information provided by patients. This implies that, for these pharmacists to deliver medication guidance, patients must not only disclose information about their medications but also divulge medical information about their illnesses.

However, when choosing a pharmacy, patients often prioritize location and short waiting times over the quality of pharmacological management [7,8,9]. Research thus far has also generally shown a discrepancy between the perceptions of patients and pharmacists about pharmacist services. In 2010, Gidman et al. [10] revealed that in the west of Scotland, patients primarily perceive the main duty of pharmacists to be supplying drugs. Meanwhile, in 2004, Worley et al. [11] reported that in the United States, patients’ perceptions significantly differ from pharmacists’ self-perceptions regarding various aspects of the roles of pharmacists in terms of “helping [patients] to fulfill their wishes regarding health maintenance and improvement”, “assisting with medication management”, “offering health support”, “assisting with medication concerns”, “listening when they have questions about their medication”, and “greeting” patients. Furthermore, in 2018, Kim et al. [12] reported that patients in South Korea were less likely to share information with pharmacists, especially regarding adverse drug reactions, drug–drug interactions, and past drug allergies. In 2011, Inoue et al. [13] revealed a comparable use of language and volume and speed of speech between pharmacists and patients; however, patients‘ recognition was lower than that of pharmacists in terms of discussing daily life, confirming side effects and laboratory data, and understanding concerns and worries. In a 2017–2018 survey, patients in Japan wanted to consult with pharmacists about self-care, whereas pharmacists believed that it was important to reduce waiting times and be courteous [14,15]. These findings indicate a gap between pharmacists’ behaviors and patients’ perceptions of the pharmacist role, as well as the need for more community pharmacists and pharmacist services.

In 2020, Kishimoto et al. [16] reported that factors that facilitated patients’ self-disclosure to pharmacists included trust in the community pharmacist, recognition of the pharmacist as a medical professional, understanding of the community pharmacist’s duties, and the presence of a family pharmacist. However, the extent to which patients perceive pharmacists’ behaviors related to information sharing and its potential impact on patients’ trust and willingness to self-disclose remains unclear.

Therefore, to improve information sharing and medication guidance behaviors of community pharmacists in Japan, this study aimed to investigate disparities between pharmacists and patients differed in their recognition of pharmacists’ behaviors related to information sharing and how patients’ recognition of pharmacists’ behaviors related to information sharing influence their trust in community pharmacists and their willingness to self-disclose.

## 2. Materials and Methods

### 2.1. Study Design, Setting, and Participants

This cross-sectional study involved patients and community pharmacists. Patients were surveyed online using stratified sampling. Respondent panels were provided by Cross Marketing Inc. to avoid the potential selection bias that could arise if pharmacists requested survey participation from patients who were visiting their pharmacies. Cross Marketing had a total of 2.95 million active panels that could be surveyed, and the survey was conducted in December 2020. Eligible respondents were those who regularly visited a healthcare provider and received prescription drugs at a pharmacy (including prescriptions for family members). Participants who were healthcare providers or had a family member who was a healthcare provider were excluded. No specific disease or prescription drug was excluded in this study. The survey was conducted using 5 age categories (≤39 years and younger, 40s, 50s, 60s, and 70–89 years) with 120 participants in each category (600 in total). Further, to eliminate the influence of gender differences, a male-to-female ratio of 1:1 was employed. The survey area was not limited in order to assess the situation in Japan as a whole.

A least 600 community pharmacists who worked full-time and had at least 1 year of experience working in the 4 largest dispensing chain pharmacies in Japan [17] (i.e., AIN Holdings Inc., Nihon Chouzai Co., Ltd., KRAFT Inc., and QOL Co., Ltd.) were selected. At least 150 participants were selected from each of these 4 companies. Participants included pharmacists working in pharmacies with which the co-researchers were associated using simple random sampling in each company. This survey was conducted from September to November 2021 using online self-administered survey tools (Questant, https://questant.jp/, (accessed on 3 September 2021), Macromill, Inc., Tokyo, Japan) and in-company systems. Each company was asked to collect data from 150 participants.

This study was approved by the Ethics Committee on Research Involving Human Subjects, Graduate School of Pharmaceutical Sciences, Showa University (approval number: 384). Consent was obtained from the research participants via a consent item in the questionnaire.

### 2.2. Sample Size

With a patient-to-pharmacist ratio of 1:1, an effect size *d* of 0.2 for differences in recognition of information sharing between patients and pharmacists as calculated from previous studies [11], an alpha error of 0.05, and a power of 80%, the number of patients required in each survey group was 552. G*Power version 3.1.9.7 [18] was used to calculate the sample size.

### 2.3. Survey Contents

The Appendix A presents the survey content. The questionnaires for pharmacists and patients are presented in Appendix A, respectively. The items common for both pharmacists and patients were recognition of pharmacists’ behaviors related to information sharing (9 items: 7-point Likert scale, from 1, “very strongly disagree”, to 7, “very strongly agree”), which in turn were based on the survey items developed by Worley et al. [11].

Only patients were asked to indicate their trust in community pharmacists (7 items: 6-point Likert scale, from 1, “very strongly disagree”, to 6, “very strongly agree”) based on the survey items in Kishimoto et al. [19], their willingness to self-disclose to pharmacists (5 items: 7-point Likert scale, from 1, “very strongly disagree”, to 7, “very strongly agree”) based on Kishimoto et al. [16], the medical conditions them (self-report), whether the family pharmacist (the same pharmacist the patient goes to) has explained them to you, the number of medications they took, and the number of pharmacies they went to regularly.

Only pharmacists were asked about their work experience (years), sex, whether they had obtained education that made them qualified by the Japan Education Pharmacists Center, and whether they had other certifications in Japan.

The survey items regarding the recognition of pharmacists’ behaviors related to information sharing by Worley et al. [11] were not developed as a scale and have not been validated. However, they have been cited in many studies; thus, we believe that their external validity is assured. “Trust in pharmacy” by Kishimoto et al. [19] and “Willingness to self-disclose” by Kishimoto et al. [16] have been validated as a scale. “Trust in pharmacy” and “Willingness to self-disclose” have also been used in a previous study [20].

### 2.4. Statistical Analyses

Nine items about pharmacists’ behaviors were compared between the two groups and analyzed using the Mann–Whitney *U* test to calculate an effect size *r* to identify differences in the recognition of pharmacists’ behaviors related to information sharing between community pharmacists and patients [21]. The associations between patients’ recognition of pharmacists’ behaviors related to information sharing and patients’ age, number of medications taken, and community pharmacists’ recognition of pharmacists’ behaviors related to information sharing and years of experience were evaluated by calculating Spearman’s correlation coefficients.

Path analysis was conducted to investigate the impact of patients’ recognition of pharmacists’ behaviors related to information sharing on patients’ trust in community pharmacists and their willingness to self-disclose. In addition, patient recognition was factor analyzed (maximum likelihood method, Promax rotation), and the number of factors was confirmed by scree plotting. The confidence coefficient, McDonald’s *ω* [22], was confirmed to exceed 0.70 and added.

SPSS Statistics version 26 (IBM Corp., Armonk, NY, USA) was used for statistical analyses, and SPSS Amos version 26 (IBM Corp.) was used for path analysis. McDonald’s *ω* was calculated using the OMEGA macro for SPSS provided by Hayes [23]. Statistical significance was set at *p* < 0.05.

## 3. Results

### 3.1. Respondent Attributes

Responses were obtained from 600 patients. In total, 50% of the patients were males, with a median age (interquartile range, IQR) of 55 (43–67) years. The most common medical conditions were, in the order of prevalence from high to low, hypertension (33.3%), dyslipidemia (20.3%), psychiatric disorders (17.5%), allergic disease (15.8%), and insomnia (10.3%). The median (IQR) of the aggregate of the 7 items of trust in pharmacists (min 6–max 42) and the aggregate of the five items of willingness to self-disclose (min 5–max 35) was 28.0 (24.0–31.0) and 20 (15.0–24.0), respectively (Table 1).

In total, 845 pharmacists totaling 4 companies were asked to complete the survey, and 645 responded. Most respondents worked in an urban area in Kantou (eastern half of Japan, including Tokyo). Of the pharmacists surveyed, 54 worked part-time and were excluded from the study (69.9% effective response rate). Table 2 shows the pharmacists’ attributes. In total, 32.8% of the community pharmacists in the sample were males. The male-to-female ratio was similar to that among all pharmacists in Japan [24].

### 3.2. Differences between Patients and Pharmacists Regarding the Recognition of Pharmacists’ Behaviors Related to Information Sharing

Recognition of pharmacists’ behaviors related to information sharing was significantly lower for patients than for pharmacists for all items, with scores of 4–5 for patients and ≥5 for pharmacists (Table 3). The effect sizes for all items were at least moderate, and the aggregate of the 9 items was as large as 0.62. Items with a medium effect size were “Talk with patients even if the patients don’t have any medication questions”, “Make sure that patients understand how to use their medications before they leave the pharmacy”, and “Be easily approachable to discuss a patient’s medication concerns.” Items with a large effect size were “Talk with patients about how to watch for medication side effects”, “Talk with patients about whether or not it is OK to take their medications with over-the-counter products”, “Show an interest in working with patients to meet their healthcare needs”, “Communicate a desire to help patients manage their medications”, and “Communicate a desire to help patients with their medication concerns”; the item with the largest effect size was “Listen to patients when they have a medication question”.

Patients’ recognition of pharmacists’ behaviors related to information sharing obtained significant correlation coefficients with patients’ age, and number of medications, which were significantly different from the explanation provided by the family pharmacists. Patients’ age obtained significant correlation coefficients in two items: “Show an interest in working with patients to meet their healthcare needs” and “Listen to patients when they have a medication-related question.” The numbers of medications obtained significant correlation coefficients in three items: “Communicate a desire to help patients manage their medications”, “Listen to patients when they have a medication question”, and “Be easily approachable to discuss a patient’s medication concerns.” The explanations provided by the family pharmacists were significantly different for all items. However, no item showed a high correlation (with a correlation coefficient *ρ* > 0.2) or a large effect size (with an effect size of *r* > 0.3); therefore, there was no relationship (Table 4).

Pharmacists’ recognition of their own behaviors related to information sharing obtained significant correlation coefficients with their years of experience which were significantly different in having one or more certifications. Their years of experience obtained significant correlation coefficients in one item: “Communicate a desire to help patients with their medication concerns.” Having one or more certifications was significantly different on one item: “Talk with patients about whether or not it is OK to take their medications with over-the-counter products.” However, no item showed a high correlation, with a correlation coefficient *ρ* > 0.2, or a large effect size (with an effect size of *r* > 0.3); therefore, there was no relationship (Table 5).

### 3.3. Patients’ Recognition of Pharmacists’ Behaviors Related to Information Sharing and Its Association with Patients’ Trust in Community Pharmacists and Willingness to Self-Disclose

Through a factor analysis, patients’ recognition of pharmacists’ behaviors related to information sharing, patients’ trust in the pharmacist, and patients’ willingness to self-disclose yielded one factor with McDonald’s *ω* of 0.940, 0.945, and 0.921, respectively, indicating good internal consistency. We envisioned a multiple regression model that predicted willingness to self-disclose based on patients’ trust in community pharmacists and patients’ recognition of pharmacists’ behaviors related to information sharing, combined with a single regression model that predicts patients’ trust in community pharmacists based on patients’ recognition of pharmacists’ behaviors related to information sharing (Figure 1).

The results of the path analysis showed that this model was saturated with zero degrees of freedom, and thus the goodness of fit could not be calculated. Therefore, the coefficient of determination, R^2^, was used as an evaluation index for this model [25]. The R^2^ value was 0.51; as this exceeds 0.5, the model is determined a good fit [26]. There was a significant and positive direct path from patients’ recognition of pharmacists’ behaviors related to information sharing to patients’ trust in community pharmacists (0.66, *p* < 0.001). There was also a significant and positive direct path from patients’ recognition of pharmacists’ behaviors related to information sharing to patients’ willingness to self-disclose (0.45, *p* < 0.001). The direct path from patients’ trust in community pharmacists to patients’ willingness to self-disclose was also significant and positive (0.34, *p* < 0.001). We also observed a significant and positive indirect path from patients’ recognition of pharmacists’ behaviors related to information sharing to patients’ willingness to self-disclose (0.22, *p* < 0.001).

## 4. Discussion

This study indicates a gap between community pharmacists and patients in the recognition of pharmacists’ behaviors related to information sharing. A previous study showed a trend that was similar to our findings, in which patients had a lower recognition of pharmacists’ behaviors than did pharmacists [11]; nevertheless, our results demonstrated a greater difference between the two groups compared with the previous study. The following six items had the most significant effect sizes: “Talk with patients about how to watch for medication side effects”, “Talk with patients about whether or not it is OK to take their medications with over-the-counter products”, “Show an interest in working with patients to meet their healthcare needs”, “Communicate a desire to help patients manage their medications”, “Communicate a desire to help patients with their medication concerns”, and “Listen to patients when they have a medication question.”

Our findings showed that pharmacists had a median score ranging from 5 to 7 regarding the recognition of pharmacists’ behaviors related to information sharing; this indicates that community pharmacists in Japan may not be clarifying these behaviors when interacting with patients. To improve pharmacist–patient communication, community pharmacists should be more assertive in clarifying their intentions.

Our findings also show that patients’ recognition of pharmacists’ behaviors related to information sharing were not statistically related to patient age, number of medications taken, or the explanation provided by the family pharmacist. Thus, pharmacists may need to clarify their intentions before providing medication guidance, regardless of the type of patient they are working with. Our findings also show that pharmacists’ recognition of their behaviors was not statistically related to years of experience or their certification. Given that pharmacists have high recognition of behavior even when they have little experience or are not certified, it is possible that they have low goals for their “must be” behavior and overestimate their own behavior. To increase the goals for their “must be” behavior, it is necessary to strengthen the pharmacists’ training or education in terms of communication and providing medication advice.

The path analysis results revealed that the patients’ recognition of pharmacists’ behaviors related to information sharing had a positive influence on patients’ trust in community pharmacists and their willingness to self-disclose. There was also a positive influence of patients’ trust in community pharmacists on their willingness to self-disclose. This indicates that pharmacists could be more proactive and engage in closer communication with their patients so as to better understand and assess the patients’ perspectives regarding the wanted and unwanted effects of medications, their adherence to medication, and living conditions. If we consider our evidence in light of the behavioral change strategies identified by Jamil et al. [27], community pharmacists engaging in reinforcing behaviors related to “Feedback and monitoring”, “Social support”, and “Shaping knowledge” could help bridge this gap between patients and community pharmacists in Japan and promote patients’ willingness to self-disclose. Prior research has identified the following factors as facilitators of pharmacist interventions with homebound patients: positive recognition of the value of the pharmacist’s role and the existence of a relationship, trust, and respect toward the pharmacist [28]. Meanwhile, Druică et al. [29] cited in their study that pharmacist attentiveness (represented by the items “The pharmacist was interested in my needs”, “The pharmacist treats all customers the same”, and “The amount of time the pharmacist spent giving me medication advice was sufficient”) is one of the factors influencing patient confidence in the information received from pharmacists. These citations and the current study’s evidence collectively show that, if pharmacists act more proactively when engaging with patients, this may be associated with patients giving more recognition to the role of pharmacists, which may then be directly and indirectly associated with an increase in patients’ willingness to self-disclose. The greater amount of information acquired through these proactive processes may also enable pharmacists to more appropriately implement medication management measures.

It has been reported that, on average, pharmacists in Japan spend about 2–5 min per patient delivering medication instructions [30]; however, this length of time is unlikely to enable a pharmacist to deliver comprehensive instruction about medications, including the specific items addressed in this study: “Talk with patients about how to watch for medication side effects”, “Talk with patients about whether or not it is OK to take their medications with over-the-counter products”, “Show an interest in working with patients to meet their healthcare needs”, “Communicate a desire to help patients manage their medications”, “Communicate a desire to help patients with their medication concerns”, and “Listen to patients when they have a medication question.” The pharmacist may reduce the amount of time they spend on patient instruction because of the patients’ time-related concerns as well as the impact of the standard of the number of community pharmacists assigned to each pharmacy. In Japan, pharmacists are responsible for all pharmacy activities, including prescriptions, dispensing medications, and providing medication guidance and documentation. As a result, it is common for Japanese patients to be kept waiting at the pharmacy. Since waiting time has been reported to affect patients’ criteria for choosing a pharmacy [7,8,9] and satisfaction with the service [29], it is possible that pharmacists themselves are overly concerned with the time they spend with patients, and this may be associated with the little time spent on medication instruction. However, the time for medication instruction itself was not reported to affect patient satisfaction. As the quality of the pharmacist’s response and medication instruction affect patient satisfaction [31], pharmacists can gain patient trust and contribute to improving patient satisfaction by spending sufficient time in providing careful medication instructions. Nonetheless, it is important to emphasize at this point that, in Japan, the minimum standard for the assignment of pharmacy pharmacists is one pharmacist for an average of 40 out-of-hospital prescriptions per day [32]. These descriptions imply that the number of pharmacists in a Japanese pharmacy tends to be low, which may not allow the pharmacists enough time to provide appropriate medication guidance to each patient. Therefore, to ensure that pharmacists have sufficient time to provide medication guidance and reduce patient waiting time, pharmacies may need to actively increase the number of pharmacists on their staff, provide pharmacists with assistants for the dispensing processes, and mechanize the pharmacy’s dispensing and auditing processes.

This study has two limitations. First, it did not examine all factors that influence patients’ trust in community pharmacists and patients’ willingness to self-disclose information to community pharmacists. Instead, this study only examined the patients’ recognition of pharmacists’ behaviors related to information sharing. Future research could examine the attitudes of pharmacists (e.g., sincerity when providing medication guidance, knowledge related to drug treatment evaluation, and counseling skills) that may influence patients’ trust in community pharmacists and willingness to self-disclose. Second, the assessment of the recognition of pharmacists’ behaviors related to information sharing may have diverged by group, as the patient and pharmacist groups were surveyed separately. However, this study included pharmacists from four of Japan’s leading chain pharmacies, and for both groups, the sample size was considered sufficient for allowing generalizations; however, the constraints of the study’s design and methodological limitations must be considered. Although there were differences in the study area, Japan is a small island country with no extreme differences in economic situation or level of education, and the number of pharmacists in pharmacies was determined by the number of prescriptions, and the amount of co-payment by patients is a fixed percentage determined by the universal health insurance system. Therefore, we believe this has little impact on the perception of behaviors by areas. Future researchers should examine the extent to which improvements in the recognition of pharmacists’ behaviors related to information sharing influence patients’ trust in community pharmacists and, thereby, their willingness to self-disclose.

## 5. Conclusions

While community pharmacists in Japan recognized pharmacists’ own behaviors related to information sharing, patients did not show a similar level of recognition, revealing a significant perception gap. This highlights that it may be important for pharmacists to engage more proactively in interactions with patients and better articulate the purpose of their actions. Medication therapy could be proactively assessed by pharmacists, which could include confirming drug effects and unwanted effects for patients and assessing patients’ medication adherence and living conditions. These actions may support the improvement of patients’ trust in and willingness to self-disclose to community pharmacists.

## Figures and Tables

**Figure 1 pharmacy-12-00063-f001:**
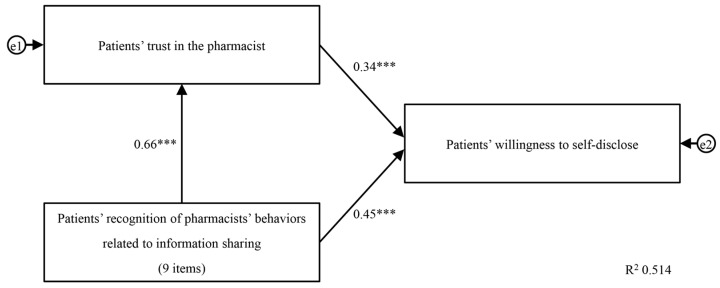
Path diagram of patients’ recognition of pharmacists’ behaviors related to information sharing. Patients’ recognition of pharmacists’ behavior related to information sharing positively relates to patients’ trust in community pharmacists and willingness to self-disclose. The numbers are standardization coefficients. e1 and e2 are standard errors. *** *p* < 0.001.

**Table 1 pharmacy-12-00063-t001:** Patients’ attributes, n = 600.

Characteristics	Values	
Age, years (median (IQR))	55.0	(43.0–67.0)
Gender (n, (%))		
Male	300	(50.0)
Female	300	(50.0)
Number of medications (median (IQR))	3.0	(2.0–5.0)
Use of more than one pharmacy (n, (%))	132	(22.0)
An explanation was provided by the family pharmacist (n, (%))	463	(77.2)
Medical condition (n, (%))		
Hypertension	200	(33.3)
Dyslipidemia	122	(20.3)
Psychiatric diseases	105	(17.5)
Allergic disease	95	(15.8)
Insomnia	62	(10.3)
Trust in pharmacist (min 6–max 42, median (IQR))	28.0	(24.0–31.0)
Willingness to self-disclose (min 5–max 35, median (IQR))	20.0	(15.0–24.0)

IQR, interquartile range. “Trust in pharmacist” was measured by a total score of 7 items (6-point Likert scale: 1, “very strongly disagree” to 6, “very strongly agree”) and “Willingness to self-disclose” by a total score of 5 items (7-point Likert scale: 1, “very strongly disagree” to 7, “very strongly agree”), with a higher number indicating greater agreement for both measurements.

**Table 2 pharmacy-12-00063-t002:** Pharmacists’ attributes, n = 591.

Characteristics	Values	
Years of experience (median (IQR))	8	(4–15)
Gender (n, (%))		
Male	194	(32.8)
Female	397	(67.2)
Certifications (n, (%))		
One or more certifications	488	(82.2)
Education system-qualified by the Japan Education Pharmacists Center	479	(81.0)
Primary care-certified pharmacist	1	(0.2)
Board-Certified Pharmacist in Home Care Pharmacy	3	(0.5)
JPEC-Certified Pharmacist in Pediatric Pharmacotherapy	2	(0.3)
Pharmacist certified in Chinese and other herbal medicines	10	(1.7)
Certified Practical Training Supervisor Pharmacist	56	(9.5)

IQR, interquartile range.

**Table 3 pharmacy-12-00063-t003:** Difference between patients and pharmacists regarding the recognition of pharmacists’ behaviors related to information sharing.

Pharmacists’ Behaviors Related to Information Sharing ^a^		Patients		Pharmacists				
		(n = 600) ^b^		(n = 591) ^b^		*p* Value ^c^	Effect Size *r*	
1	Talk with patients about how to watch for medication side effects.	4	(3–5)	6	(5–6)	<0.001	0.53	Large
2	Talk with patients even if the patients do not have any medication questions.	5	(4–5)	6	(5–6)	<0.001	0.41	Medium
3	Talk with patients about whether it is OK to take their medications with over-the-counter products.	4	(3–5)	6	(5–6)	<0.001	0.53	Large
4	Show an interest in working with patients to meet their healthcare needs.	4	(3–5)	6	(5–6)	<0.001	0.51	Large
5	Communicate a desire to help patients manage their medications.	4	(3–5)	5	(5–6)	<0.001	0.53	Large
6	Make sure that patients understand how to use their medications before they leave the pharmacy.	4	(4–5)	6	(5–6)	<0.001	0.48	Medium
7	Communicate a desire to help patients with their medication concerns.	4	(3–5)	6	(5–6)	<0.001	0.59	Large
8	Listen to patients when they have a medication question.	5	(4–6)	7	(6–7)	<0.001	0.64	Large
9	Be easily approachable to discuss a patient’s medication concerns.	5	(4–6)	6	(5–6)	<0.001	0.39	Medium
Total score (9 items)		39	(33–45)	52	(47–55)	<0.001	0.62	Large

^a^ 7-point Likert scale from 1, “very strongly disagree”, to 7, “very strongly agree.”; ^b^ median (IQR), and IQR stands for interquartile range; ^c^ Mann–Whitney *U* test.

**Table 4 pharmacy-12-00063-t004:** Relationship of patients’ recognition of pharmacists’ behaviors related to information sharing with other items (n = 600).

		Age		Number of Medications		Explanation Provided by the Family Pharmacist ^a^	
Patients’ Recognition of Pharmacists’ Behaviors Related to Information Sharing		Spearman’s Correlation Coefficient *ρ*		Spearman’s Correlation Coefficient *ρ*		Effect Size *r*	
1	Talk with patients about how to watch for medication side effects.	0.00		0.04		0.15	***
2	Talk with patients even if the patients do not have any medication questions.	0.04		0.04		0.18	***
3	Talk with patients about whether it is OK to take their medications with over-the-counter products.	−0.05		0.00		0.17	***
4	Show an interest in working with patients to meet their healthcare needs.	0.11	**	0.08		0.17	***
5	Communicate a desire to help patients manage their medications.	0.07		0.10	*	0.20	***
6	Make sure that patients understand how to use their medications before they leave the pharmacy.	0.01		0.00		0.20	***
7	Communicate a desire to help patients with their medication concerns.	0.05		0.07		0.16	***
8	Listen to patients when they have a medication question.	0.09	*	0.09	*	0.21	***
9	Be easily approachable to discuss a patient’s medication concerns.	0.08		0.11	**	0.20	***
Total score (9 items)		0.06		0.08		0.22	***

* *p* < 0.05, ** *p* < 0.01, *** *p* < 0.001. ^a^ Mann–Whitney *U* test.

**Table 5 pharmacy-12-00063-t005:** Relationship with pharmacists’ recognition of pharmacists’ responses (n = 591).

		Years of Experience		Having One or More Certifications ^a^	
Pharmacists’ Recognition of Pharmacists’ Behaviors Related to Information Sharing		Spearman’s Correlation Coefficient *ρ*		Effect Size *r*	
1	Talk with patients about how to watch for medication side effects.	−0.03		0.08	
2	Talk with patients even if the patients do not have any medication questions.	0.00		0.03	
3	Talk with patients about whether it is OK to take their medications with over-the-counter products.	−0.03		0.10	*
4	Show an interest in working with patients to meet their healthcare needs.	−0.07		0.04	
5	Communicate a desire to help patients manage their medications.	−0.05		0.08	
6	Make sure that patients understand how to use their medications before they leave the pharmacy.	−0.01		0.03	
7	Communicate a desire to help patients with their medication concerns.	−0.09	*	0.04	
8	Listen to patients when they have a medication question.	−0.07		0.05	
9	Be easily approachable to discuss a patient’s medication concerns.	0.00		0.04	
Total score (9 items)		−0.05		0.07	

* *p* < 0.05. ^a^ Mann–Whitney *U* test.

## Data Availability

Ethical approval was contingent upon all study data being stored electronically on an external storage disk. The datasets used and analyzed in this study are available from the corresponding author upon reasonable request.

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
