# Peer review of "Recognition of Community Pharmacists’ Behaviors Related to Information Sharing: A Cross-Sectional Study"

_pharmacy, 2024, doi:10.3390/pharmacy12020063_

Round 1

Reviewer 1 Report

Comments and Suggestions for Authors

First of all, I would like to congratulate the authors of the article, they wrote an interesting article which shows relevant information that could improve pharmacy practice and professional development. I would like to make some recommendations for improvement of the article.

I recommend adding more information about participants, from which areas of Japan were particpants? and what are the characteristics of those areas? urban areas, rural areas, sociodemographic characteristics, ..?

I would like to know if the authors can obtain the data of how many people were asked to participate and what is the percentage of participation, and if it is possible, what was the reason for non-participation.

In the description of the simple size it is not clear which variable the size effect refers to.

Is the Worley et al. developed survey validated? If so, please indicate it in the article. If it is not a validated survey, I consider that it is a limiting factor to be included in the discussion.

In Table 3, correct in the last line of the total score the range of patient scores.

I wonder if the researchers have considered as a limitation the fact of not knowing the sociodemographic characteristics that might affect the responses.

Author Response

Thank you very much for reviewing our manuscript. We have summarised our responses to your comments in the attached file, please see the attachment.

Reviewer 2 Report

Comments and Suggestions for Authors

Authors state to explore a critical step in pharmacy care – the patient pharmacist trust interaction with an emphasis on information sharing from a large diverse sample; as suggested by the title.  They are working to advance pharmacy practice in Japan, which is laudable.  The study is confusing because I thought it would be mostly about the pharmacist patient relationship that creates trust and willingness of the sample to share health information.   Most of the study is on whether the pharmacist performs aspects of medication evaluation and education with patients and if the patient agrees the pharmacist performs this service. The pharmacists say they mostly do while the patients say sometimes they do but perception is always lower than the pharmacists' self-assessment. Most of the sample has a family pharmacist (463/600), which would skew the few trust and willingness to share parameters measured I would assume to more trust.  The study tries to find correlations with pharmacists providing clinical pharmacy services but important aspects as staffing, time, privacy, duration of person as a client, training, etc were not assessed, only years of experience and certifications (very low number of pharmacists).  Some of this briefly discussed in the limitaitons.  I think the paper needs to be rewritten and discuss primary findings in terms of pharmacists' clinical responsibilities from patient and pharmacist perspectives with a secondary goal of trust and willingness to share personal health information for the subgroup that doesn’t have a pharmacist family member 923% of sample). This would involve two study objectives and title change.   Overall the study is well written but has much duplication between table/figure and text.  Generally data are presented in only one format. Some terminology for measurement appears to mean different things to authors vs. this reviewer.  The journal editors can determine if this is an issue or not.

Major

Ln 77 – correct verb appears to be identify not improve.  This study did not work on improving perceptions, practices, and beliefs between pharmacists and patients.   Half of the study evaluates pharmacist performance with the last quarter on trust and willingness to share.   Might be better to list objectives as two fold to better characterize the study data.

Ln 93 – appears a biased sample to measure trust when 77% of the patient sample has a pharmacist as a family member.  This should be used as a confounder in the data analysis, however the sample with a family pharmacist now becomes very small 23%.   Or only use the data from the sample with a pharmacist is analyzed, greatly reducing your power below your power calculation.

Ln 105 – please describe how the companies selected pharmacists to understand if any bias at this step.

Ln 122 – too bad this wasn’t converted to a 7 point scale to easily compare to the other results. Need to mention this problem every time you present trust data so readers do not assume the upper limit of the item was 7

Ln 140 – need to discuss the creation of a sum score for this analysis

Ln 154-162 - almost all this information is in table 1.  Please delete from text and just refer to the table for attributes

Ln 163 – trust was on a scale of 1 – 6, so not clear what these numbers are.  Why not median and IQR?

Ln 163 – willingness to share was on a scale of 1 – 7, so not clear what these numbers are.  Why not median and IQR?

Ln 164 – 167 – almost all of this information is in table 2. Please delete from text and just refer to the table for attributes

Ln 187 – Title of the table does not truly reflect the data.   It seems the patients are asked about tasks the pharmacists does, not their willingness to share info about their personal selves.  Most of the questions are focused on what questions or information the pharmacist provides outside of dispensing activities.   For example how well a pharmacist relays their desire to the patient to help does not measure if the patient is willing to share personal medical and medication information.

Table 4 – not clear what the explanation provided by the family pharmacist analysis is measuring.  Does this measure if the family pharmacist provided this information instead of their regular pharmacist?

Figure 1 and text 233-244 basically say the same things, duplication.  However, unclear on how many readers know how to understand path analyses.

Ln 338 – need to discuss the major confounder in 77% of patients who have a family pharmacist, which directly influences trust and willingness to share.

Minor

Ln 53 - provide country for the study

Ln 54 - provide country for the study

Ln 60 - provide country for the study

Ln 62- you have dates for a few studies but not all.  Would either give dates for all or none

Ln 62 - this is unclear on what was compared or what was expected, please revise.

Ln 64 - not clear what lower recognition is, please revise

Ln 66 - provide country for the study

Ln 93 - states participants excluded if a family member was a health care provider, but 77% of the sample had a pharmacist as a health care provider.  Do you not consider a pharmacist a health care provider in Japan?  

Ln 128 – 130 – is a certificate the same as a license in other countries?  What other certifications would they have?

Ln 158 – diseases in order of prevalence, not importance

Ln 163 – change title to Patients’ attributes

Ln 163 – might be better to state the percent that used more than one pharmacy as your median and IQRs are all 1.

Ln 164 – in the footnote, state that trust was measured on a 6 point Likert scale and willingness to self-disclose on a 7 point Likert scale, with higher the number the greater the agreement for both measurements

Ln 163 – how do you get so many participants to have a family member who is a pharmacist?  This high percentage greatly influences your trust results.

Ln 169 – change to Pharmacists’ attributes

Ln 187 – need a line space before total score

Ln 228 – Shouldn’t the e1 and e2 be values vs circles?

Ln 233 – how does one determine the model was saturated with zero degrees of freedom?

Ln 246 – how do you determine substantial? 2 point difference in perceptions?

Ln 247 – seems the pharmacist is not doing recognition but validating if they provide these clinical skills.  The questions ask if you agree pharmacists do x, y and z, not if you feel these steps are important or welcomed by the patient.

Ln 257 define who they are

Ln 263 – this is more of a hypothesis than a conclusion from your study.   You did not measure value or patient desire to receive this care only if pharmacists have provided assessment and medication education.

Ln 268 – suspect bigger problems than awareness.  Your framework was very limited to understand practice and reasons not performing these steps.

Ln 282 – isn’t this pharmacist behavior change?  What behavior does the patient need to change? Ask the pharmacist for these services?   You don’t measure any patient behaviors.

Ln 292 – you didn’t ask if the patients valued any of the clinical medication services pharmacists can provide.  Recognition isn’t value, appreciation, etc.   Maybe some language differences between countries in terms of word meanings.  

Comments on the Quality of English Language

Some terms like recognition and information sharing seem to have different definitions between authors and reviewer. 

Author Response

Thank you for reviewing our manuscript. We have summarised our responses to your comments in the attached file, please see the attachment.

Reviewer 3 Report

Comments and Suggestions for Authors

The article deals with pharmacy services include the effective assessment of medication therapy, consideration of side effects, and improvement of medication adherence this research investigated the following topics: whether pharmacists and patients differed in their recognition of pharmacists’ behaviors related to information sharing; whether and how patients’ recognition of pharmacists’ behaviors related to information sharing may influence patients’ trust in community pharmacists and their willingness to self-disclose

The introduction clarifies the reason for the study, highlighting the growing role of community pharmacy in health care

The methods are very explicit and appropriate, the statistical methodologies are well described. The tables are clear and relevant and the figures contribute to better understanding

The discussion is appropriate and relevant, the analysis of the relationship between effectiveness and efficiency is interesting and opens doors for other studies. Reflection on the limitations of the realistic study

Considering the need for community pharmacy to play a more significant role in healthcare, this work brings together relevant and meaningful information obtained from users

Author Response

Thank you for reviewing our manuscript. 
In this study, we asked pharmacists and patients the same qusestions. In practice, the questions were fine-tuned to suit the concerns of patients and pharmacists. However, the actual questions were not shown in this study, which made some of the questions difficult to understand. Therefore, the questionnaires used have been added as supplementary material.
We hope you will find it useful.

Reviewer 4 Report

Comments and Suggestions for Authors

Thank you for the opportunity to review this paper. I found it well-written, interesting and well-designed. I have not detected any issues that would require further clarification from the authors and I think the paper can be accepted as it is. 

Author Response

(The authors gave the same response as above.)
